# Personal and Organizational Factors as Predictors of Life Satisfaction Among Older Adults in Long-Term Care Settings

**DOI:** 10.3390/healthcare13030306

**Published:** 2025-02-02

**Authors:** Xiaoli Li, Cheng Yin, Juliana O. Abude-Aribo, Reagen Conklin, Elias Mpofu

**Affiliations:** 1School of Health Sciences, Southern Illinois University, Carbondale, IL 62901, USA; reagen.conklin@siu.edu; 2Rehabilitation and Health Services, University of Norh Texas, Denton, TX 76203, USA; chengyinunt@gmail.com (C.Y.); umaabude-aribo@my.unt.edu (J.O.A.-A.); elias.mpofu@unt.edu (E.M.); 3School of Health Sciences, University of Sydney, Sydney, NSW 2006, Australia; 4Educational Psychology, University of Johannesburg, Johannesburg P.O. Box 524, South Africa

**Keywords:** personal factors, organizational factors, satisfaction predictors, long-term care

## Abstract

Background and Aims: Resident satisfaction is a critical indicator of the quality of care in long-term care facilities (LTCFs). Yet, the relative importance quality of care factors in predicting resident satisfaction remains unclear for guiding resident support initiatives. This study aims to identify the relative contribution of personal and facility care service factors as predictors of satisfaction LTCF residents. Methods: This cross-sectional study enrolled a convenience sample of 399 older adult residents from LTCFs in Shanghai, Nanjing, and Changsha, China, from June to December 2023. The inclusion criteria were age 65 or older, fluency in speaking and reading simplified Chinese Mandarin, having resided in long-term care facilities for at least one month, and cognitive competence to comprehend the questionnaires. Hierarchical multiple regressions were utilized to examine self-report survey data on ten resident personal variables, and seven care facility service domains were examined for their relative contribution to resident care satisfaction. Moreover, the analysis included the interaction between personal factors and care service attributes. Results: The combined personal and care service factors explained 26.1% of the variance in overall resident satisfaction. Personal factors that predicted resident satisfaction included age, level of independence, and length of stay (ΔR^2^ = 0.11). Of the care facility factors, the significant predictors of higher resident satisfaction were spending time (β = 0.60, *p* < 0.01, ΔR^2^ = 0.09) and environment domains (β = 0.62, *p* < 0.01, ΔR^2^ = 0.03). Age moderated the relationship between the spending time domain and overall satisfaction, with a positive effect for residents aged 70–79 compared to those aged 60–69 (β = −1.26, *p* < 0.05). Conclusions: This study provides evidence to suggest the importance of personal and care facility characteristics to LTFC resident satisfaction. Based on these findings, improved resident satisfaction is likely with LTCF care services that provide tailored care plans using resident characteristics.

## 1. Introduction

The demand for long-term care (LTC) services continues to grow globally due to increased life expectancy and the expanding population of older adults. Long-term care facilities (LTCFs) provide comprehensive personal, social, and medical services and support to older adults to maintain or recover their functional ability consistent with their basic rights and human dignity [1]. They comprise settings, such as nursing homes, assisted living, and memory care centers [2]. Despite the critical role of resident satisfaction as a quality metric reflecting how well providers meet residents’ values and expectations [3,4,5,6], the advancement of LTC models and satisfaction assessment tools in non-western countries remains limited. We aimed to address these gaps in the evidence, investigating personal and LTCF organizational factors associated with resident satisfaction in the Chinese context.

The role and functions of LTCFs: LTC systems are vital for supporting individuals with significant capacity declines, ensuring they live with dignity and exercise their rights and freedoms. They vary widely in the services they provide, which would be particularly the case in developing country settings with a shorter, emerging history providing such services. Effective long-term care encompasses a range of services, from managing chronic conditions to providing assistive care and social support, all delivered within an integrated and person-centered framework [7]. They are less well studied in developing country settings, where older adult care services are still emerging. Conceivably, the needs that LTCF would meet vary with the individual characteristics of the residents as well as care facility environment factors.

Personal factors in LTCF resident satisfaction: Personal factors refer to the background of an individual’s life and living, including features of the individual that are not part of a health condition or health states and which can impact functioning positively or negatively [8]. These factors encompass gender, race, age, other health conditions, fitness, lifestyle, habits, upbringing, coping styles, social background, education, profession, past and current experience, overall behavior pattern and character style, individual psychological assets, and other characteristics. [9]. Personal factors associated with LFTC satisfaction, such as better health status [10], female gender [11], higher autonomy [12], and more frequent visits from friends and family [13], are positively associated with satisfaction in LTCFs. For instance, residents in better physical and mental health often report higher satisfaction LTCFs due to their increased ability to engage with their surroundings and care routines [10], with a sense of autonomy. Conceivably, residents who have autonomy or greater ability to make decisions about their daily lives and care tend to report higher levels of satisfaction. However, autonomy may vary in meaning by cultural context, of which collectivist cultures may value interdependence more than sense of individualistic control of social outcomes. Family relationships would be important to how residents of LTFC regard their satisfaction with life regarding the frequency of family and social visits [14,15].

Organizational factors and resident satisfaction: Previous studies on resident satisfaction in LTCFs identified key organizational factors that contribute to higher levels of satisfaction. Generally, organizational factors in LTCFs include staffing levels, management style, training programs, quality improvement initiatives, resident assessment processes, communication practices, safety culture, regulatory compliance, financial stability, facility layout, and the overall organizational structure, and these significantly impact the quality of care provided to residents and the well-being of staff within the facility [16]. For instance, environmental factors such as the physical layout of the facility and the availability of communal spaces impact resident social interactions and satisfaction [17]. However, staff shortages and high workloads may limit the ability to maintain consistent, meaningful interactions. In developing countries, LTCFs may focus on the quality of basic care and resource availability leaving certain organizational factors less explored. Evidence on the influence of a clean, safe, and esthetically pleasing environment is less conclusive in developing countries despite being appreciated by some residents. Menn et al. [18] highlight that improving the environment of LTCFs can heighten satisfaction, while the practical implementation of resident-centered LTFCs remains to be studied for implementation in developing countries.

The primary aim of this study is to identify the personal and resident care predictors of resident satisfaction in Chinese LTCFs. The United Nations projects that 16% of the global population will be over 65 years of age by 2050 [19]. China is no exception, experiencing a rapid growth in its aging population aged 60 and older, which has nearly doubled to approximately 255 million people in the past two decades [20]. In addition, we explore the moderating effects of personal factors, including age cohort, level of independence, and length of stay, on the relationship between organizational service domains and overall resident satisfaction. We addressed the following research questions regarding resident satisfaction in LTCFs in the Chinese context:What is the relative contribution of personal and care facility factors to LTCF resident satisfaction?How does the interaction of personal and care facility factors explain LTCF resident satisfaction?

## 2. Materials and Methods

### 2.1. Study Design

A cross-sectional survey was conducted to assess the validation of the Chinese version OLCRSS in long-term care facilities. This study was approved by institutional review boards (IRB) of the University of North Texas.

### 2.2. Participants and Setting

This study enrolled a convenience sample of 399 older adult residents from LTCFs in Shanghai, Nanjing, and Changsha, China, from June to December 2023. The sample included a mix of residents with varying levels of independence, including 40% who were classified as independent and 48% who were married. Unlike LTCFs in countries such as Sweden and the United States, where admission is often restricted to older adults with significant frailty or care needs, LTCFs in China accommodate individuals who are relatively younger, healthier, and seeking convenience or social engagement rather than medical or functional support [21]. This cultural and systemic difference in LTCF utilization reflects the unique social and familial structures in China, where older adults may choose LTCFs to alleviate familial caregiving burdens or to access better living conditions.

The inclusion criteria were as follows: aged 65 or older, fluency in speaking and reading simplified Chinese Mandarin, having resided in long-term care facilities for at least one month, and cognitive competence to comprehend the questionnaires. The exclusion criteria included residents with severe cognitive impairments or dementia as determined by their medical records or caregiver assessments, those with significant hearing or speech impairments that prevented effective communication, and individuals with terminal illnesses or other conditions that might preclude participation in this study.

G*Power 3.1 software was used to ensure sufficient power to detect significant effects [22]. The analysis was based on the inclusion of 10 tested predictors in the hierarchical multiple regression model, a medium effect size (f^2^ = 0.15), and a power level of 0.80, as recommended by Cohen [23]. This calculation indicated a minimum required sample size of 119 participants. The final sample of 326 participants exceeded the minimum requirement and aligns with general guidelines for regression analyses, supporting the adequacy of our sample for this analysis.

### 2.3. Measures

#### 2.3.1. Resident Satisfaction Measure

Resident satisfaction was assessed using the Chinese version of the Ohio Long-Term Care Resident Satisfaction Survey (OLCRSS), a validated tool designed to evaluate satisfaction across multiple domains of care. The OLCRSS has demonstrated strong psychometric properties, including a content validity index (CVI) of 1.0, intraclass correlation coefficient (ICC) of 0.96 (*p* < 0.001), and Cronbach’s alpha of 0.96. The total satisfaction score, serving as the outcome variable, was measured on a continuous scale ranging from 0 to 100.

#### 2.3.2. Predictor Variables

Personal factors: Gender was categorized as a dichotomous variable, with male coded as (1) and female as the comparison group (0). Age was divided into four cohorts, 60–69, 70–79, 80–89, and 90+, and treated as a categorical variable in the analysis. Education level was categorized into four groups: primary school, middle school, high school, and bachelor’s degree or higher. Marital status was categorized into four groups: married, single, divorced, and widowed. Living arrangement was categorized into three groups: separate room, twin room, and multiple-bedded room. Level of independence was grouped into three categories: independent, half-independent, and completely dependent. The chronic disease count was categorized as 1, 2, or 3 or more.

Organizational/care facility factors: The Chinese OLCRSS questionnaire included seven domains: moving in, spending time, care and services, caregivers, meals and dining, environment, and facility culture. Each domain consisted of several items scored on a 5-point Likert scale, where higher scores indicated greater satisfaction, with no coded as 1, probably no as 2, neutral as 3, probably yes as 4, and yes as 5 (see Table 1). For example, the moving in domain included items such as orientation help and warm welcome, while the spending time domain included meaningful activities and community connection. There are two variables (waiting time and staff anger) that were initially coded, with higher scores indicating lower satisfaction. To facilitate its reporting, both variables were reverse-coded in the subsequent analysis so that higher scores indicated greater satisfaction.

### 2.4. Procedure

This study was approved by the Institutional Review Boards (IRB) of University of North Texas, IRB-21-250. All participants provided written informed consent before enrolling in the study. The consent form detailed the study’s purpose, procedures, potential risks and benefits, data usage, and participants’ rights. Participants were explicitly informed that their data would be used for research purposes and that they could withdraw from the study at any time without penalty. The consent process was conducted securely via a protected platform. Following this procedure, 326 older adult residents were recruited and provided valid data for the study.

### 2.5. Data Analysis

The data analysis was conducted in two stages: descriptive and inferential.

Descriptive Analysis

Initially, we analyzed the data derived from the demographic characteristics of the participants and their association with the residents’ total satisfaction scores. Frequencies and percentages were calculated for categorical variables (e.g., gender, age cohort, and living arrangement), while means and standard deviations were reported for continuous variables (e.g., satisfaction score).

Inferential Analysis

*t*-tests and ANOVAs were performed to assess the relationships between demographic variables and overall satisfaction. To better represent the contribution of each domain to the total satisfaction score, linear regression models were used to assign weights to the individual items within each domain. This approach provided a more precise calculation of how much variance in total satisfaction could be explained by each domain [24,25,26,27]. For instance, the results of the composite spending time domain explained 10% of the variance in total satisfaction, which more accurately represented the original explained variance (see Appendix A). By assigning weights to each item, the contribution of each domain to overall satisfaction was more precisely reflected, enhancing the robustness of the findings. A similar analysis was conducted for the remaining six domains.

A hierarchical multiple regression analysis was performed to assess the impact of various satisfaction domains on predicting residents’ overall satisfaction in LTCFs. All assumptions for multiple regression were met [28]. Resident demographics, including age, gender, living arrangement, level of independence, chronic conditions, and length of stay in nursing homes, were identified as potential factors influencing overall satisfaction, based on the results of *t*-tests and ANOVA. These factors were controlled in the first step of the regression analysis. In the subsequent step, each of the seven composite satisfaction domains (e.g., moving in, spending time, and care and services) was added, resulting in a total of eight regression models.

To examine the moderating effects of personal factors on the relationship between the spending time domain and overall satisfaction, interaction terms were created for each moderator. For example, age was categorized into four groups: 60–69 years (reference group), 70–79 years, 80–89 years, and 90+ years. We applied Model 1 of the PROCESS macro (version 4.2) in SPSS to assess its interactions.

All statistical analyses were performed using SPSS (version 28), and a *p*-value of <0.05 was considered statistically significant. The overall confidence level for this study was 95%.

## 3. Results

### 3.1. Descriptive Statistics

Table 1 presents the descriptive statistics for the study variables. Age was significantly linked to satisfaction, with residents aged 70–79 reporting the highest scores (Mean = 91.45) and those aged 90+ the lowest (Mean = 88.12; *p* < 0.05). Males had higher satisfaction levels compared to females (Mean = 89.85 vs. 89.16, *p* < 0.01). Residents in separate rooms reported greater satisfaction than those in twin or multiple-bedded rooms (Mean = 91.90 vs. 90.24 and 87.45, respectively, *p* < 0.01). Completely dependent residents had the highest satisfaction (Mean = 92.57), while fewer chronic conditions were associated with higher satisfaction (Mean = 89.66, *p* < 0.01). Regarding length of stay in nursing homes, residents with over 3 years of residence reported the highest satisfaction (Mean = 91.62), while those with less than 2 months of residence reported the lowest satisfaction (Mean = 86.53, *p* < 0.05).

### 3.2. Composite Satisfaction Domain Correlations

The intercorrelations among variables (Table 2) show that six composite satisfaction domains are significantly positively related to overall resident satisfaction, except for the composite moving in domain. The relationships between these composite satisfaction domains and total satisfaction are statistically significant, but the effect sizes of these correlations range from small to medium [29].

### 3.3. Personal Factor Predictors

The hierarchical regression (Table 3) shows that, when combined, all predictor variables accounted for 26.1% of the variance in total resident satisfaction. In Step 1, those six resident demographics (age, gender, living arrangement, level of independence, chronic conditions, and length of stay in nursing homes) were entered and significantly contributed to overall satisfaction, accounting for 11.4% of the variance. In the subsequent steps, each composite satisfaction domain was added individually.  

### 3.4. Organizational Attributes Predictors

For example, in Step 2, the composite moving in domain was introduced, but it did not significantly improve the model (ΔR^2^ = 0.003). In Step 3, the composite spending time domain was added, which explained an additional 8.7% of the variance. In the final step, after all seven composite domains were included, the model accounted for 26.1% of the variance in total resident satisfaction, with the most significant contribution from the composite spending time domain (β = 0.20, *p* < 0.01, ΔR^2^ = 0.09) and followed by the composite environment domain (β = 0.18, *p* < 0.01, ΔR^2^ = 0.029). In addition, demographics including age (β = −0.11, *p* < 0.05), level of independence (β = 0.24, *p* < 0.01), and length of stay (β = 0.13, *p* < 0.01) were significant contributors in the final model.

### 3.5. Moderation of Age, Level of Independence, and Length of Stay

Given the minimal impact of the environmental domain (R^2^ = 0.03) in the hierarchical regression, we excluded this predictor in the moderation analysis to maintain clear and meaningful insights.

After adding the three moderators in the relationship between the spending time domain and overall satisfaction, only one significant interaction was found. This interaction was between the spending time domain and the age cohort of 70–79 years compared to the reference group (60–69 years) (β = −1.26, *p* < 0.05). The overall model explained a total of 17% of the variance in overall satisfaction (R^2^ = 0.17). No other significant interactions or moderators were found (See Table 4).

## 4. Discussion

In this study, the satisfaction of residents in LTCFs and the factors related to this were explored to provide foundational data for nursing interventions aimed at improving satisfaction in this population. Among the organizational factors, the composite spending time domain and the composite environment domain were the most significant contributors. Individual factors, such as age, level of independence, and length of stay, also played crucial roles in shaping satisfaction levels.

### 4.1. Overall Satisfaction

Our findings also reveal that residents in LTCFs reported high overall satisfaction scores, ranging from 86.53 to 92.57 out of 100, consistent with previous studies [30,31,32]. This high satisfaction could be attributed to the relatively early stage of development of the LTC system in China, where a limited number of older adults currently access these services. As education and economic levels of users and their families rise, the standards of care they expect and receive are likely to increase. Additionally, convenience sampling was employed due to its practicality and accessibility, selecting participants who were readily available and met the inclusion criteria (see Appendix A). Consequently, the sample may have included more residents in LTC facilities who had a better health status and who were less dependent. This reflects a unique characteristic of Chinese LTC facilities, which often admit independent individuals seeking not just care, but also a more engaging community environment [33]. These residents may have rated their satisfaction levels higher as a result.

While users reported high levels of overall satisfaction, it is crucial to identify and understand the specific factors contributing to their satisfaction. This understanding will help guide targeted improvements in LTC services and interventions.

### 4.2. Personal Factors

Resident satisfaction showed a declining trend with increasing age. This decline suggests that long-term care quality may not fully meet the needs of older residents as they age. The growing older adult population, coupled with inadequate care resources to address their needs, is a pressing concern in long-term care settings. Similar patterns of decreasing satisfaction with age have been observed in studies conducted in China [34], Italy [35], and Korea [36]. However, other research, such as Chou’s study, showed an opposite trend, with increasing life satisfaction reported as individuals age [37]. Additionally, some studies found no significant relationship between resident age and satisfaction components in long-term care settings [29]. These varying findings suggest that older adults may adapt to their circumstances over time and that their satisfaction can increase if healthcare providers effectively address their needs.

This adaptability aligns with two other individual predictors—level of independence and length of stay—which were significant contributors to resident satisfaction in the final model. Residents who stay in long-term care facilities for more than a year may become more familiar with their environment and feel more comfortable in it. This familiarity likely contributes to higher satisfaction with the quality of care, even as their dependence on caregivers increases. Previous studies also support this finding, highlighting that accessibility to high-quality care services enables residents to better utilize these services and benefit their overall health status [38].

Given the physical vulnerabilities of older adults, their reliance on nearby and accessible services is understandable. These findings emphasize that policies aimed at ensuring adequate care resources and high-quality care delivery are crucial. Such policies can support the provision of resident-focused care, ultimately improving satisfaction levels among long-term care residents.

### 4.3. Organizational Factors

Among the seven organizational predictors of resident satisfaction, the spending time domain and the environment domain emerged as the strongest contributors. The spending time domain includes factors such as time enjoyment, daily anticipation, community connection, meaningful activities, special events, activity preferences, and weekend activities. Given the relatively isolated social conditions in residential care homes, group recreational activities within and outside the facilities, along with fostering social connections and engagement, are key areas for improving resident satisfaction. Previous studies on organizational factors influencing satisfaction have demonstrated that physical activity, social recreational activities, and greater engagement are positively associated with higher levels of resident satisfaction [39,40,41].

In LTC settings, the opportunities for interaction and involvement in community activities can have a profound impact. Social factors, such as participation in meaningful activities and forming connections, often outweigh the influence of medical care and professional treatment in enhancing life satisfaction [42]. To improve resident satisfaction, it is important to encourage residents to perform the activities of daily living to the greatest extent possible. Providing appropriate skills, knowledge, guidance, and emotional support can help manage loneliness and depression among residents. Building a robust social support network and strengthening organizational support should be prioritized.

The environment domain includes aspects such as cleanliness, room navigation, outdoor access, privacy, and safety. While environmental comfort factors like cleanliness, room navigation, and outdoor access are foundational characteristics of a satisfactory physical environment, this study highlights the importance of privacy and safety for nursing home residents. These findings align with previous research, which has shown that easy access to outdoor spaces and green areas promotes social interaction among residents, ultimately enhancing satisfaction [43,44,45,46]. Furthermore, Lee et al. [47] found that residents with physical disabilities are more likely to communicate with nurses and prioritize safety when walking. In shared living spaces, where many residents live in shared rooms, concerns about privacy and the safety of personal belongings significantly impact their quality of life. Thus, ensuring security and maintaining privacy are critical factors in fostering a sense of satisfaction among residents. These findings underline the importance of designing physical and social environments in long-term care facilities that prioritize residents’ comfort, safety, and opportunities for meaningful engagement.

The findings of this research identified several predictors of residents’ satisfaction from both organizational and resident perspectives. If policymakers aim to use resident satisfaction as a metric for quality monitoring or pay-for-performance initiatives, it is crucial to focus on these key predictors. For example, tailored approaches that account for age-related needs and provide additional support for more dependent individuals can improve their experiences. On the organizational level, strategies that prioritize meaningful activities, social engagement, and a well-designed accessible environment can foster a sense of comfort, security, and connection. These efforts can significantly contribute to improved mental health, social well-being, and overall satisfaction among residents in Chinese long-term care settings [48]. Policymakers and care providers can use these findings to inform quality improvement initiatives, policy development, and resource allocation, ensuring that care delivery in long-term care facilities meets the complex and evolving needs of residents.

In addition, the moderation analysis provided valuable insights into the relationship between the spending time domain and overall satisfaction. Notably, age emerged as a significant moderator, with the 70–79 age cohort showing a negative interaction effect compared to the 60–69 reference group. This finding suggests that spending time may influence satisfaction differently across age groups, potentially reflecting varying preferences or needs related to social engagement or time utilization among older residents. By addressing both individual and organizational factors, long-term care facilities can create environments that support residents’ well-being and satisfaction, ultimately enhancing the quality of care provided.

### 4.4. Limitations and Future Directions

This study was conducted in Shanghai, Nanjing, and Changsha, areas where long-term care services are more developed compared to other regions in China, particularly rural areas. As a result, the findings may not be fully generalizable to long-term care facilities in other locations. Future studies with larger and more diverse resident samples are recommended to further determine the satisfaction indicators and assess variations in quality across different types of facilities. The cross-sectional design of this research presents a limitation, as it captures data at a single point in time, which restricts the ability to observe changes or trends over time. Conducting longitudinal research in future could help establish causal relationships and track changes over time, providing a deeper understanding of the dynamics between variables. Despite these limitations, our findings can help health policymakers and the management of LTCFs set priorities in improving LTC services based on their residents’ care needs.

## 5. Implications for Research and Practice

Our findings highlight the importance of both individual and organizational factors in shaping resident satisfaction in LTCFs. Studies show that person-centered care models, where facilities adapt to the unique preferences and routines of residents, lead to higher satisfaction and improved quality of life [49]. Positive interactions between staff and residents, marked by empathy, active listening, and responsiveness, are critical for fostering trust and emotional well-being [50]. Future research should explore these dimensions, including how technological innovations like telehealth or assistive devices could enhance person–LTCF interactions while maintaining a human-centered approach.

Social engagement, environmental comfort, and personal independence underscore determinants of satisfaction, while tailored care approaches are necessary to address unique needs. Additionally, the experiences of residents can be enhanced by fostering a sense of belonging and community. As LTC systems in developing regions continue to evolve, high-quality standards and adaptation of services to meet the increasing expectations of residents and their families will be non-negotiable.

## 6. Conclusions

This study provides valuable insights into the factors influencing resident satisfaction in LTCFs. Both resident-specific characteristics, such as age, level of independence, and length of stay, were strong predictors of LTCF residents’ overall satisfaction. Organizational factors, including the spending time domain and the environment domain, emerged as significant predictors of resident overall satisfaction. The interaction between age and the spending time domain showed a less pronounced positive association for residents aged 70–79 compared to those aged 60–69. These findings highlight the importance of personal and care facility environment factors for initiatives to enhance the overall quality of care and life satisfaction for residents in long-term care facilities in a collectivistic culture country setting.

## Figures and Tables

**Table 1 healthcare-13-00306-t001:** Descriptive Demographic characteristics of the participants and their relationship with resident total satisfaction scores (n = 326).

Variable	n (%)	Resident Satisfaction Scores		
		Mean	SD	t/F
Gender				7.34 **
Male	111 (34)	89.85	7.47	
Female	215 (66)	89.16	8.50	
Age cohort				3.65 *
60–69	34 (10.4)	91.30	5.82	
70–79	80 (24.5)	91.45	6.76	
80–89	162 (49.7)	88.38	8.36	
90+	50 (15.3)	88.12	10.05	
Education level				1.04
Primary school	166 (50.9)	89.80	8.32	
Middle school	101 (31)	88.24	7.88	
High school	52 (16)	90.17	8.34	
Bachelor’s degree or more	7 (2.1)	90.83	6.47	
Marriage status				1.56
Married	155 (47.5)	89.98	8.01	
Single	52 (16)	89.79	9.50	
Divorce	23 (7.1)	90.71	7.37	
Widowed	96 (29.4)	87.92	7.70	
Living arrangement				6.50 **
Separate room	41 (12.6)	91.90	5.01	
Twin room	162 (49.7)	90.24	7.33	
Multiple bedded room	123 (37.7)	87.45	9.56	
Level of independence				4.42 *
Independent	130 (39.9)	88.25	8.19	
Half independent	156 (47.9)	89.54	8.02	
Completely dependent	40 (12.3)	92.57	7.92	
Chronic diseases count				5.57 **
1	96 (29.4)	91.50	6.92	
2	164 (50.3)	88.06	8.36	
3 or more	66 (20.2)	89.66	8.75	
Daily activity participation				0.40
1	108 (33.1)	89.75	7.59	
2	169 (51.8)	89.01	8.33	
3 or more	49 (15)	89.96	8.86	
Monthly expenditures				2.03
0–999	7 (2.1)	87.38	3.25	
1000–2999	123 (37.7)	90.79	6.64	
3000–4999	94 (28.8)	88.79	8.43	
5000+	102 (31.3)	89.40	8.16	
Length of stay in nursing homes				2.63 *
0–2 months	31 (9.6)	86.53	7.04	
3–12 months	103 (31.6)	88.61	8.37	
1–3 years	168 (51.5)	90.09	8.14	
3 years or more	24 (7.4)	91.62	7.85	

t = *t*-test; F = ANOVA; SD = standard deviation; * *p* < 0.05; ** *p* < 0.01.

**Table 2 healthcare-13-00306-t002:** Intercorrelations of composited domain predictors and outcome variables.

Variable	1	2	3	4	5	6	7	8
1 Total satisfaction score	-	-	-	-	-	-	-	-
2 Composite moving in	0.090	-	-	-	-	-	-	-
3 Composite spending time	0.337 **	0.041	-	-	-	-	-	-
4 Composite care and services	0.265 **	−0.039	0.350 **	-	-	-	-	-
5 Composite caregivers	0.321 **	−0.037	0.484 **	0.609 **	-	-	-	-
6 Composite meals and dining	0.273 **	−0.077	0.356 **	0.581 **	0.579 **	-	-	-
7 Composite environment	0.299 **	−0.119 *	0.277 **	0.458 **	0.386 **	0.422 **	-	-
8 Composite facility culture	0.266 **	−0.027	0.336 **	0.589 **	0.556 **	0.493 **	0.512 **	-

* *p* < 0.05; ** *p* < 0.01.

**Table 3 healthcare-13-00306-t003:** Summary of hierarchical regression for predictors of total satisfaction in nursing homes (*n* = 326).

Variable	*B*	*SE B*	β	R^2^	ΔR^2^
Step 1				0.114	0.114
Age cohort	−1.497	0.542	−0.157 **		
Living arrangement	−1.777	0.706	−0.145 *		
Level of independence	2.583	0.677	0.212 **		
Length of stay in nursing homes	1.493	0.574	0.140 *		
Step 2				0.116	0.003
Age cohort	−1.593	0.551	−0.167 **		
Living arrangement	−1.914	0.720	−0.156 **		
Level of independence	2.765	0.702	0.227 **		
Length of stay in nursing homes	1.516	0.574	0.142 **		
Moving in	−0.635	0.574	0.142		
Step 3				0.203	0.087
Age cohort	−1.117	0.530	−0.117 *		
Level of independence	2.878	0.668	0.236 **		
Chronic diseases count	−1.390	0.656	−0.119 *		
Length of stay in nursing homes	1.343	0.547	0.126 *		
Moving in	−0.567	0.623	−0.051		
Spending time	0.908	0.154	0.306 **		
Step 4				0.218	0.014
Level of independence	2.852	0.663	0.234 **		
Chronic diseases count	−1.442	0.651	−0.124 *		
Length of stay in nursing homes	1.216	0.545	0.114 *		
Moving in	−0.367	0.624	−0.033		
Spending time	0.793	0.161	0.267 **		
Care and services	0.506	0.209	0.134 *		
Step 5				0.226	0.008
Age cohort	−1.037	0.526	−0.108 *		
Level of independence	2.731	0.664	0.224 **		
Length of stay in nursing homes	1.284	0.545	0.120 *		
Moving in	−0.232	0.626	−0.021		
Spending time	0.675	0.173	0.227 **		
Care and services	0.275	0.244	0.073		
Caregivers	0.388	0.216	0.125		
Step 6				0.230	0.004
Age cohort	−1.073	0.526	−0.112 *		
Level of independence	2.701	0.664	0.221 **		
Length of stay in nursing homes	1.311	0.545	0.123 *		
Moving in	−0.155	0.628	−0.014		
Spending time	0.650	0.174	0.219 **		
Care and services	0.151	0.261	0.040		
Caregivers	0.308	0.224	0.099		
Meal and dining	0.323	0.241	0.088		
Step 7				0.258	0.029
Age cohort	−1.033	0.517	−0.108 *		
Level of independence	2.842	0.653	0.233 **		
Length of stay in nursing homes	1.420	0.536	0.133 **		
Moving in	0.094	0.622	0.008		
Spending time	0.595	0.171	0.200 **		
Care and services	−0.069	0.264	−0.018		
Caregivers	0.279	0.220	0.090		
Meal and dining	0.197	0.240	0.054		
Environment	0.671	0.192	0.201 **		
Step 8				0.261	0.002
Age cohort	−1.039	0.518	−0.109 *		
Level of independence	2.941	0.662	0.241 **		
Length of stay in nursing homes	1.409	0.536	0.132 **		
Moving in	0.038	0.625	0.003		
Spending time	0.595	0.171	0.200 **		
Care and services	−0.126	0.272	−0.033		
Caregivers	0.228	0.227	0.073		
Meal and dining	0.172	0.242	0.047		
Environment	0.616	0.201	0.184 **		
Culture	0.238	0.257	0.063		

* *p* < 0.05; ** *p* < 0.01; Only significant demographic variables are reported in this table. Step 1 included six variables: gender, age cohort, living arrangement, independence level, chronic disease count, and length of stay. Step 2 = Step 1 + composite moving in variable; Step 3 = Step 2 + composite spending time variable; Step 4 = Step 3 + composite care and services variable; Step 5 = Step 4 + composite caregiver variable; Step 6 = Step 5 + composite meals and dining variable; Step 7 = Step 6 + composite environment variable; Step 8 = Step 7 + composite facility culture variable.

**Table 4 healthcare-13-00306-t004:** The interaction between spending time and moderators on total satisfaction among LTCF residents.

Predictors	β	SE	95% CI
Spending Time and Age			
Spending time	1.16 **	0.45	0.27–2.05
Spending time × (Age 70–79 vs. 60–69)	−1.26 *	0.55	−2.14–0.02
Spending time × (Age 80–89 vs. 60–69)	0.27	0.52	−0.76–1.30
Spending time × (Age 90+ vs. 60–69)	−0.01	0.54	−1.07–1.05
Spending Time and Level of Independence			
Predictors	β	SE	95% CI
Spending time	1.10 **	0.26	0.58–1.61
Spending time × (half independent vs. independent)	−0.09	0.35	−0.78–0.60
Spending time × (completely dependent vs. independent)	−0.12	0.42	−0.94–0.70
Spending Time and Length of Stay			
Predictors	β	SE	95% CI
Spending time	0.78	0.53	−0.27–1.82
Spending time × (3–12 months vs. 0–2 months)	0.53	0.60	−0.65–1.71
Spending time × (1–3 years vs. 0–2 months)	0.22	0.58	−0.92–1.35
Spending time × (3 years or more vs. 0–2 months)	−0.44	0.72	−1.85–0.98

* *p* < 0.05; ** *p* < 0.01; LTCF = long-term care facility. The models show that the spending time domain, along with the interaction terms for different moderators, explain a notable proportion of the variance in total satisfaction: 17% for age (R^2^ = 0.17), 15% for level of independence (R^2^ = 0.15), and 14% for length of stay (R^2^ = 0.14).

## Data Availability

Dataset available on request from the authors.

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
