# Peer review of "Personal and Organizational Factors as Predictors of Life Satisfaction Among Older Adults in Long-Term Care Settings"

_healthcare, 2025, doi:10.3390/healthcare13030306_

Round 1
Reviewer 1 Report
Comments and Suggestions for Authors
Dear authors,
thank you very much for the opportunity to read this study, which aims to identify relative contribution of personal and facility care service factors as predictors of satisfaction LTCFs residents. This is an observational study that must comply with the STROBE guideline.
Abstract: The abstract should provide more information about the method (in addition to the design and population) including setting, type of sampling, inclusion and exclusion criteria, analysis process. Do not add results on the population included in the methods sub-section.
Keywords should, as far as possible, conform to MeSH terms.
The whole introduction and/or justification should be described and the objectives should be left at the end. In my opinion, there should not be a new sub-section for aims at the end of the introduction.
Check section headings to adjusts them into line to the journal standards (e.g. Material and methods).
Methods:
The first sub-heading of Material and methods should be Design.
Exclusion criteria should be more clearly defined. It would be appropriate to describe and justify a sample size estimate based on criteria of statistical precision and expected proportion or mean in the population.
In the Data analysis section, the analysis process should be described in an orderly manner: descriptive, inferential. (The results should be presented in the same order). Specify the statistical tests used. Indicate the confidence level for the study (95% CI?).
Discussion: Review the sub-sections in the discussion (personal factors, organisational factors...).
I recommend ending the discussion with limitations. This is followed by a new section with implications for clinical practice, recommendations for the future. Finally, end with conclusions.
References: check the link to the primary source or DOI where applicable.
Author Response
|
Reviewer 1 |
||
|
Comment |
Author Response |
|
|
1 |
Abstract: The abstract should provide more information about the method (in addition to the design and population) including setting, type of sampling, inclusion and exclusion criteria, analysis process. Do not add results on the population included in the methods sub-section. |
We now included the inclusion and exclusion criteria, analysis process in Abstract. We deleted the population data in the methods and added it in the results (line 15-24).
|
|
2 |
Keywords should, as far as possible, conform to MeSH terms. |
We have revised the keywords (line 34-35). |
|
3 |
The whole introduction and/or justification should be described, and the objectives should be left at the end. In my opinion, there should not be a new sub-section for aims at the end of the introduction. |
We now deleted the sub-section for aims and left the research questions at the end of Introduction (line 93-105)
|
|
4 |
Check section headings to adjusts them into line to the journal standards (e.g. Material and methods). |
We now corrected the section headings to meet the journal standards such as 2. Materials and Methods (line 106)
|
|
5 |
Exclusion criteria should be more clearly defined.
It would be appropriate to describe and justify a sample size estimate based on criteria of statistical precision and expected proportion or mean in the population. |
We have added specific information about exclusion criteria in Participants and Settings (line 120-124).
We conducted G*Power analysis to ensure sufficient power to detect significant effects. We have now added one paragraph to describe and justify a sample size estimate based on the results of the analysis (line 125-131)
|
|
6 |
In the Data analysis section, the analysis process should be described in an orderly manner: descriptive, inferential. (The results should be presented in the same order). Specify the statistical tests used. Indicate the confidence level for the study (95% CI?). |
We now added the descriptive analysis and re-ordered the analytical methods in Data Analysis and Results (line 171-178 and line 208-219)
We elaborated on the significance of the 95% confidence interval (CI) for the study (line 203-205, 208-219). |
|
7 |
Discussion: Review the sub-sections in the discussion (personal factors, organizational factors...).
I recommend ending the discussion with limitations. This is followed by a new section with implications for clinical practice, recommendations for the future. Finally, end with conclusions. |
We have revised the discussion part to include the limitation of research (line 379), and this is followed by section 5 of Implication for research and practice (line 392)
|
|
8 |
References: check the link to the primary source or DOI where applicable. |
We have checked and re-do the references format and added the link to the primary source or DOI (line 431-520) |
Reviewer 2 Report
Comments and Suggestions for Authors As to the scientific and for the authors>>>
In your Methodology you just mention a "convenience sample", might be good with a few words more on that. I ask because your Table 1 reports that 40 % of the residents are "independent" and 48 % are married. I have visited some LTCF in China and was always surprised to meet so many older people who were just old (not veryold) but not frail or in need of help. For them it was just convenient, I understood. They would not be allowed to enter a LTCF in my country (Sweden), nor would they want to. They would be much too young and healthy, wouldn't get even Home Help here... In the 1940s we had a similar situation, when older people with no place to live or in very substandard housing entered what we called old-age homes (LTCF at that time). Mostly poor also. Rich old people had other accomodations. I notice that some of the participants in your study pay several thousand yuan/month...
These facts deserve a short discussion, both in the Methods section and in the Conclusion, I believe.
Your statistical analysis seems OK to me, and it is understandable that you can explain only about a qurter of the variations, as 9/10 say they are satisfied with their care etc. And with a small standard deviation. My experience from field work and other studies is that people express more dissatisfaction in good LTCFs than in poorer/bad ones... (maybe they dare not express any complaints?).
If you attend to these queeries I believe your study will be better and raise more interest.
You also need some competent English speaker to scrutinize your language and grammar, which is problematic in a number of places. For example: line 27 "the important" > "the importance", 29 "the" DELETE, 42 "associations" >> associated, 43 don't understand the sentence, 47 & 51 repetitive, 69-70 don't understand. I am also surprised to learn in Table 1 that 48 % are "independent" - why are they in a LTCF?!
Problems with English, see above
Author Response
|
Reviewer 2 |
||
|
In your Methodology you just mention a "convenience sample", might be good with a few words more on that. I ask because your Table 1 reports that 40 % of the residents are "independent" and 48 % are married. I have visited some LTCF in China and was always surprised to meet so many older people who were just old (not veryold) but not frail or in need of help. For them it was just convenient, I understood. They would not be allowed to enter a LTCF in my country (Sweden), nor would they want to. They would be much too young and healthy, wouldn't get even Home Help here... In the 1940s we had a similar situation, when older people with no place to live or in very substandard housing entered what we called old-age homes (LTCF at that time). Mostly poor also. Rich old people had other accommodations. I notice that some of the participants in your study pay several thousand yuan/month... If you attend to these queries I believe your study will be better and raise more interest. |
We added a paragraph in Materials and Methods to explain why 40% of the residents in the research population are independent and to describe the differences in resident characteristics among China, Sweden, and the United States (109-117). The discussion now addresses the convenience sample issues explicitly, with explanation on the reasons why independent individuals go to long-term care facilities in China and the potential factors that they rated the higher satisfaction scores (line 287-296) |
|
|
You also need some competent English speaker to scrutinize your language and grammar, which is problematic in a number of places. For example: line 27 "the important" > "the importance", 29 "the" DELETE, 42 "associations" >> associated, 43 don't understand the sentence, 47 & 51 repetitive, 69-70 don't understand. |
Our co-authors reviewed and scrutinized the manuscript, we corrected all errors (line 31, 33, 55, 58, 72, 74…) |
|
Round 2
Reviewer 1 Report
Comments and Suggestions for Authors
Dear authors, thank you very much for the implementations made in the manuscript.
The abstract has been improved in this new version. There is no need to start the results section with the following sentence: ‘After applying the inclusion criteria and excluding cases with missing data’ (start directly with a description of the achieved sample).
Keywords: value switching Long-Term Care Settings to Long-Term Care (MeSH).
Material and methods: start with a first sub-heading Design (cross sectional study...)
Data analysis: the following sentence should be described in the inferential analysis section: "T-tests and ANOVA were performed to assess the relationships between demographic variables and overall satisfaction (see Table 1)"; in fact, the information on T-test and ANOVA is duplicated again in lines 193-194 (inferential analysis). Table 1 should not be cited in the methodology as it corresponds to results.
Results: As I said for the abstract section, you should not start describing the results with the sentence: "After applying the inclusion criteria and excluding cases with missing data". Such information should be indicated in the methods section.
In the discussion, results should not be repeated (e.g. 26.1% satisfaction). What should be done is to discuss the results with those obtained in other studies in the literature, in the same and in other international contexts, citing the references.
There is a lot of discussion (whole paragraphs without citations) that requires citations in the text to justify it (in the following sections: 4.1 overall satisfaction; 4.2 personal factors; 4.3 organisational factors).
In the sections on limitations and implications for practice, consider whether it is necessary to include some relevant quotations to support the claims made.
Author Response
Comments and Suggestions for Authors
Dear authors, thank you very much for the implementations made in the manuscript.
The abstract has been improved in this new version. There is no need to start the results section with the following sentence: ‘After applying the inclusion criteria and excluding cases with missing data’ (start directly with a description of the achieved sample).
Thank you, we have deleted the sentence from the Results (line 22).
Keywords: value switching Long-Term Care Settings to Long-Term Care (MeSH).
Yes, we corrected it (line 32).
Material and methods: start with a first sub-heading Design (cross sectional study...)
We have added the Study design as sub-heading 2.1. under Materials and Methods (line 104-107)
Data analysis: the following sentence should be described in the inferential analysis section: "T-tests and ANOVA were performed to assess the relationships between demographic variables and overall satisfaction (see Table 1)"; in fact, the information on T-test and ANOVA is duplicated again in lines 193-194 (inferential analysis). Table 1 should not be cited in the methodology as it corresponds to results.
We deleted the Table 1 in Data analysis (line 178-179).
We added “T-tests and ANOVA were performed to assess the relationships between demographic variables and overall satisfaction” following Data analysis (line 180-182) and deleted the duplication (line180-181).
Results: As I said for the abstract section, you should not start describing the results with the sentence: "After applying the inclusion criteria and excluding cases with missing data". Such information should be indicated in the methods section.
The sentence has been removed from the Results (line 210-211)
In the discussion, results should not be repeated (e.g. 26.1% satisfaction).
The repeated data was removed from the Discussion (line 275-276)
There is a lot of discussion (whole paragraphs without citations) that requires citations in the text to justify it (in the following sections: 4.1 overall satisfaction; 4.2 personal factors; 4.3 organisational factors).
We have added additional research papers in Discussion part (line 393, 337, 348, 365)
In the sections on limitations and implications for practice, consider whether it is necessary to include some relevant quotations to support the claims made.
We added two citations in implication for practice to support the recommendations we suggested (line 396, 398).
Reviewer 2 Report
Comments and Suggestions for Authors
Hello again,
and thanks for the improvements: Have just two remaining complaints that you may want to consider.
Line 31 should be "the importance" (not "that importance")
Line 57 suggest "as well as with care---" (delete "the")
Line 111 suggest "such as Sweden and the United States"
Comments on the Quality of English LanguageYour English is acceptable, but might be improved. Read good novels, not scientific stuff, if you want to improve your English... (a good piece of advice that I myself once got). To learn some English is easy, to use it well and elegant is hard. Maybe true of every language... Good Luck!
Author Response
Comments and Suggestions for Authors
Hello again,
and thanks for the improvements: Have just two remaining complaints that you may want to consider.
Line 31 should be "the importance" (not "that importance")
Thank you, we corrected it now (line 29).
Line 57 suggest "as well as with care---" (delete "the")
We deleted “the” (line 54).
Line 111 suggest "such as Sweden and the United States"
We have revised it (line 112), thank you!